# Comparative Analysis of Stress and Deformation between One-Fenced and Three-Fenced Dental Implants Using Finite Element Analysis

**DOI:** 10.3390/jcm10173986

**Published:** 2021-09-02

**Authors:** Chia-Hsuan Lee, Arvind Mukundan, Szu-Chien Chang, Yin-Lai Wang, Shu-Hao Lu, Yu-Cheng Huang, Hsiang-Chen Wang

**Affiliations:** 1Department of Dentistry, Kaohsiung Armed Forces General Hospital, 2, Zhongzheng 1st. Rd., Kaohsiung City 80284, Taiwan; chla27@yahoo.com.tw (C.-H.L.); chien@hotmail.com (S.-C.C.); 2Department of Mechanical Engineering, Advanced Institute of Manufacturing with High Tech Innovations (AIM-HI), and Center for Innovative Research on Aging Society (CIRAS), National Chung Cheng University, 168, University Rd., Min Hsiung, Chiayi 62102, Taiwan; d09420003@ccu.edu.tw; 3Gentle Dental Clinic, No. 400, Baotai Rd., Fengshan Dist., Kaohsiung City 80284, Taiwan; dentowanglai@yahoo.com.tw; 4Topology Medical Consultant Co., 12F., No. 812, Xinsheng Rd., Qianzhen Dist., Kaohsiung City 80284, Taiwan; topology.med@gmail.com

**Keywords:** All-on-4, finite element analysis, cancellous bone, cortical bone, von mises stress, static structural deformation

## Abstract

Finite element analysis (FEA) has always been an important tool in studying the influences of stress and deformation due to various loads on implants to the surrounding jaws. This study assessed the influence of two different types of dental implant model on stress dissipation in adjoining jaws and on the implant itself by utilizing FEA. This analysis aimed to examine the effects of increasing the number of fences along the implant and to compare the resulting stress distribution and deformation with surrounding bones. When a vertical force of 100 N was applied, the largest displacements found in the three-fenced and single-fenced models were 1.7469 and 2.5267, respectively, showing a drop of 30.8623%. The maximum stress found in the three-fenced and one-fenced models was 13.518 and 22.365 MPa, respectively, showing a drop of 39.557%. Moreover, when an oblique force at 35° was applied, a significant increase in deformation and stress was observed. However, the three-fenced model still had less stress and deformation compared with the single-fenced model. The FEA results suggested that as the number of fences increases, the stress dissipation increases, whereas deformation decreases considerably.

## 1. Introduction

Finite element analysis (FEA) is acknowledged as a theoretical solution used in industrial problems [1]. It is also used in the biomechanical study of dental implants. Techniques for assessing the influence of stress surrounding the implant structure include photoelasticity, FEA, and strain magnitude. FEA presents numerous benefits, such as precise design illustration of multifaceted models, effortless model adjustment, and ability to simulate various case scenarios by adjusting the boundary parameters in the FEM software. The genesis of osseointegration as a concept was introduced by Per-Ingvar Branemark (1969), a professor at the Institute of Applied Biotechnology, University of Goteborg. Branemark performed a series of in vivo studies on bone marrow and joint tissue with particular emphasis on tissue reaction to various kinds of injury: mechanical, thermal, chemical, and rheologic. These studies in the early 1960 s strongly suggested the possibility of osseointegration [2]. A general clinical jawbone augmentation procedure is usually essential to attain appropriate bone provision for positioning a standard implant [3]. One specific model is called the All-on-4™ concept [4,5]. The principle of this concept is to use four implants. Two implants are positioned axially in the frontal portion of edentulous jaws, whereas two other implants are distally angled to minimize the cantilever length, which permits prostheses for a maximum of 12 teeth [6,7]. The ultimate implant can be fixed or temporary. Regardless of the high success rate of these specific implants, many studies showed many important mechanical complications that are largely associated with extreme occlusal pressure and implant design [8,9,10]. The osseointegrated dental implant, which is a replacement for natural teeth, is subjected to similar stationary and dynamic forces constantly. However, the distribution of various forces to the bones through an implant is completely different as the periodontal ligament behaves as a transitional shield part [11]. Various forces are transferred to the adjoining bones directly when using a dental implant. The direct transfer of forces might instigate microfractures at the juncture of the bone and the implant, a rupture in the implant, laxation of various parts of the implant, and undesirable bone reabsorption [12]. Many kinds of research on deformation, stress, and strain distribution on many industrial implants have been conducted [13,14]. The most crucial element for the success of the implant is the fashion in which stresses are transmitted to the adjacent bone. Many key mechanical factors determine the load transfer onto adjacent bones, such as the material, quality, length, diameter, and structure of the implant [15]. FEA enables the examination and calculation of the stress dissipation efficiency of the implant [16,17]. A traditional type of implant uses only one fence along the structure of the implant [18]. The deformation and stress dissipation in traditional methods are partly responsible for the collapse of the implant [19]. However, instead of using only one fence, reinforcing the structure with more fences increases stress dissipation efficiency and decreases deformation. Therefore, this study aimed to estimate the influence of increasing the number of fences along with the structure on the stress distribution and deformation reduction on dental implants by utilizing FEA.

## 2. Materials and Methods

Several elements placed together constitute the success of a dental implant. One such element is the biocompatibility with living tissues [20,21]. Biocompatibility involves studying the physical, chemical, and mechanical properties of the material; the purpose in which the material is used; and function of the material [22,23]. In this study, biocompatibility was assessed by examining the interactions between the implant and tissues, which provide the assessment of the degree of osseointegration. For long-lasting implants, the variables that need to be considered before the design of dental implants are geometry, structure, material, surface characteristics, medical status of the patient, and bone quality [24,25].

### 2.1. Selection of the Implant Material

The biomaterials that can be used for dental implants include various metals, ceramics, polymer, and a combination of all these materials. Polymeric materials are more flexible than other biomaterials, but the usage of the former in dentistry is comparatively less than that of the latter [26]. Polyether ether ketone is a semi-crystalline polymer that has high elastic modulus. However, dental implants made from polymers have not been clinically investigated yet. Thus, animal trials must be conducted [27,28]. Ceramic implants demonstrate good bioactivity, high Young’s modulus, high brittleness, and corrosion resistance [29]. However, the major drawback of using zirconia implants is the low-temperature material degradation, which eventually results in the reduction of material strength, density, and toughness [30]. Despite clinical studies showing the short-term success of zirconia implants, long-term studies are lacking [31]. Many titanium alloys are the first-choice material for dental implants across the globe [32]. Titanium is a lightweight metal with good biocompatibility, high stiffness, and corrosion resistance. Ti-6Al-4V is the foremost material used in dental implants [33]. The failure rate of this type of dental implant is as low as 5%, and almost all implants remain in place throughout the patient’s life. Hence, in this study, the titanium alloy Ti-6Al-4V was used to analyze the one-fenced and three-fenced models at different loading conditions.

### 2.2. Geometry

Many subtle changes in geometry, such as length, diameter, and structure, will affect the failure rate of implants due to stress distribution. Research showed that a high implant length results in high stability because the contact point of the implant with the bone is high [34,35]. The length varies between 6 mm and 20 mm, and the average is between 8 mm and 15 mm [36]. In our study, 8 mm was chosen as the length of the implant, 1 mm longer than short implants, which are the favorable prosthetic solution in the bone area [37]. The radius of an implant, which is calculated as the widest spot in the implant, is given importance because of its stress dissipation [38]. A wide implant diameter results in a large amount of stress dissipation, which has been proven in animal studies. Concerning dental implants, the diameter varies between 3 and 7 mm, but the diameter is chosen depending on the bone quality, which varies among patients [17,39]. Hence, in this study, we chose an average diameter of 4.5 mm. Another factor that needs to be considered in designing a dental implant is shape. The implant can be categorized on the basis of shape, such as cylindrical, stepped, screw, conical, and hollow cylinder. The shape plays a crucial factor in deciding how the implant interacts with the bone and how forces are distributed [22]. Previous research suggested that irregular shapes, such as conical and stepped, have considerably larger stress values than smooth shapes, such as cylindrical [35]. For this specific reason, we used the plain cylindrical implant.

### 2.3. Health Condition of the Patient

Although the geometry, length, and diameter are important elements in the success of an implant, another important factor is the health status of the patient. Many clinical trials proved that the success rate of the implant is directly related to the health condition of the patient [40]. Conditions that affect the success of implants include smoking, endocrine disease, microbial and immunoinflammatory factors, cardiovascular disease, myocardial infarction, cerebrovascular accident, severe bleeding issues, and chemotherapy [41]. In a healthy patient, the failure rate is between 5% and 10% [42].

### 2.4. Standard of the Bone

One of the factors that contribute to 10% of the aforementioned failure rate is associated with poor bone quality [43]. A high bone density results in high solidity of the implant. Bone quality is a combination of skeletal size, trabecular orientation, and bone density. X-rays are used to evaluate the bone quality before the design of implant CT scans [44].

### 2.5. Design and FEA

The CAD design of the model was completed on a personal computer by using the Dassault Systemes Computer-Aided Three-Dimensional Interactive Application software (Dassault Systèmes, CATIA version 5.20, Vélizy-Villacoublay, France), whereas the analysis was completed in Ansys, version 2020 R1 (Ansys, Inc., Canonsburg, Pennsylvania, United States of America). Given the complex geometries of jawbones, the design used was simplified in the range that did not disturb the local stress analysis. Nonetheless, the design offers a base for the comparative assessment of the solutions obtained from analyses. The whole structure was made up of titanium alloy, Ti-6Al-4V (National Chung Cheng University, Chiayi County, Taiwan). Two models simulating an implant with a maximum diameter of 4.5 mm and a total length of 8 mm were developed to investigate the stress and deformation caused by load. A masticatory force was applied using a force of 100 N in a natural orientation along the *x*-axis. A titanium bar was modeled in accordance with the curvature of the implant. The radius of the bottom two fences was 2 mm, whereas the top fence had a diameter of 1.5 mm. The fence was modeled following the position of implants. The jawbone designed in this analysis was presumed as a homogeneous solid bone because the sole objective of this study was to compare the deformation and stress dissipation of both models. The two components of this study (i.e., the implant and the bone) were assumed to be isotropous, linearly elastic, and homogeneous. The mechanical properties of the materials used in this study are given in Table 1. Considering that stress changed drastically at the edges, junctures were meshed thrice as dense compared with other locals. The modeled design was a section of edentulous, posterior mandibular bone in accordance with the categorization of Lekholm and Zarb [45]. The bone adjoining the implant was designed to be 11 mm tall and 30 mm wide. The top layer in contact with the implant was the cortical bone, which was designed to be 1 mm thick, whereas the remainder constituted the trabecular bone. The average thickness of the mesh was 0.0005 mm, resulting in a total of 54,000 nodes and 18,000 elements.

## 3. Results

The results of the deformation and stress are given in Table 2 and Table 3, respectively. Figure 1 shows the comparison of static structural deformation between single-fenced and three-fenced structures, while Figure 2 shows the comparison of static structural equivalent von Mises stress between single-fenced and three-fenced structures. However, the von Mises stress found in the three-fenced model was significantly less than that in the one-fenced model (see Appendix A for the equivalent von Mises stress when using the three-fenced and one-fenced implants, respectively). Generally, the results showed that as the number of fences increased, the stress and deformation were significantly reduced because the amount of area directly in contact with the implant increased with increasing number of fences. This additional area in contact with the implant helped in dissipating stresses to the fences. With three fences at 12 contact points, the stress dissipated efficiently compared with only four contact points in one fence. The stress dissipated in one such contact point is shown in Figure 3 (see Appendix A for a close view of the transmission of stress when using the three-fenced and one-fenced implants, respectively). The largest displacements of 1.7469 and 2.5267 µm were found at the point of the load on the top of the three- and one-fenced models, respectively, showing a drop of 30.8623% (see Appendix A for total deformation when using the three-fenced and one-fenced implants, respectively). However, the highest stresses found in the three- and one-fenced models were 13.518 and 22.365 MPa, respectively, showing a drop of 39.557%.

However, the same design was also tested at an angular force of 100 N at 35° along the *x*-axis. The results of the deformation and stress are given in Table 4 and Table 5, respectively. Similar to the vertical loads, the von Mises stress found in the three-fenced model was significantly less than that in the one-fenced model (see Supplementary, Videos S7 and S8 for the equivalent von Mises stress when using the three-fenced and one-fenced implants, respectively, for an angular load of 100 N). The maximum deformation found in the one-fenced model was 171.95 µm, while that in the three-fenced model was 169.04 µm (see Appendix A for the total deformation of the three-fenced and one-fenced models, respectively, for an angular load of 100 N). The maximum stress found in the three-fenced model was 245.34 MPa, whereas that in the single-fenced model was 320.7 MPa. A significant reduction in stress by 85 MPa and maximum deformation by 3 µm was observed. Despite the significant increase in deformation and stress from a vertical load while an oblique load is applied, the three-fenced model still had less stress and deformation compared with the single-fenced model.

Mastication is the process in which food is crushed and mixed with saliva to form a bolus for swallowing. It is a complex mechanism involving the opening and closing of the jaw, secretion of saliva, and mixing of food with the tongue [46]. The occlusal biting load in the posterior jaw is usually about three times that found in the anterior jaw [47]. Hence, the same model was tested by increasing the vertical load on the posterior jaw to 150 N while decreasing the load on the anterior jaw to 50 N. The results of the deformation and stress are given in Table 6 and Table 7 respectively. The maximum deformation found in the one-fenced model was 6.8344 µm, whereas that in the three-fenced model was 5.366 µm (see Appendix A for the total deformation of the three-fenced and one-fenced models when increasing the vertical load on the posterior jaw to 150 N while decreasing the load on the anterior jaw to 50 N, respectively). The maximum stress found in the three-fenced model was 40.963 MPa, whereas that in the single-fenced model was 64.467 MPa (see Appendix A for the equivalent von Mises stress of the three-fenced and one-fenced models when increasing the vertical load on the posterior jaw to 150 N while decreasing the load on the anterior jaw to 50 N, respectively). The average stress dissipation of the three-fenced model was 0.925 MPa, whereas that in the one-fenced model was 0.94536 MPa. This finding shows that the stress dissipation was better and more evenly distributed in the three-fenced model than in the one-fenced model.

## 4. Discussion

This design follows the concept of All-on-4, which uses only four implants in the frontal portion of edentulous jaws. Two implants are placed axially, and two other implants are loaded in a distal angle to decrease the cantilever length for an immediately loaded prosthesis. The primary aim of designing a dental implant is to achieve ideal stress distribution at levels of the whole superstructure and implant infrastructure. The deformation and stress distribution vary on the basis of the design, material bone quality, and structure. One case study does not give conclusive results, and two designs are subjected to different magnitudes of force from 10 N to 150 N. All case studies have similar results. The reductions in deformation and stress are constant at 30.8623% and 39.557%, respectively, at various loads (Figure 4 and Figure 5, respectively). This study proves that the deformation and stress can be decreased remarkably by using three reinforced fences instead of one fence. However, computer-controlled simulations do not include all the variables involved in real-life situations. The same design is manufactured using titanium properties listed in Table 1 for further testing of the mode (Figure 6). Nevertheless, this design for implants in real-life situations is not motionless and causes small movements, thereby loosening the implant. A provisional restoration can vastly decrease the discomfort experienced by patients. However, the drop in the stress and deformation is evident as the contact area increases with the number of fences, which helps in dissipating stress efficiently. One particular design cannot determine the reduction in stress and deformation as the geometry, distance between each implant, standard of the bone, and health condition will vary with each patient. However, this study will help us understand that the three-fenced model has better stress dissipation efficiency than the traditional one-fenced model. The results showed that when the number of fences is increased to three, the stress and deformation are significantly reduced as the amount of area directly in contact with the implant increases with increasing number of fences. This extra area in contact with the surface of the implant helps in dissipating stresses to the fences. With three fences at 12 contact points, the stress dissipated efficiently compared with only four contact points in one fence. Another factor that can affect the titanium implants due to the stress is fatigue [48,49]. However, a recent study using commercial purity titanium of grade 1 after processing by equal-channel angular pressing (ECAP) showed that the elongation to failure is reduced due to the low hardening capability. Processing by ECAP increases the fatigue life of real implants subjected to cyclic bending loads when using the dental implant testing standard [50].

## 5. Conclusions

Modern dental implants were first used almost half a century ago, and many kinds of research have been conducted to improve them. In recent years, the affordability, efficacy, and comfort of dental implants have remarkably improved. Many parameters were considered and evaluated before designing. Many factors, such as the material used, design, behavior, geometry, and structure of the implant; bone quality; and health conditions of the patient, varied with each patient, thereby influencing the success of the implant over time. However, the results of this study revealed that this design significantly reduced the deformation and stress by using metal-reinforced provisional restoration. Using three fences instead of one fence reduced the deformation by 74% and the stress by 80%, which were detected using FEA. As such, many clinical studies should be conducted to confirm the results obtained in the FEM.

## Figures and Tables

**Figure 1 jcm-10-03986-f001:**
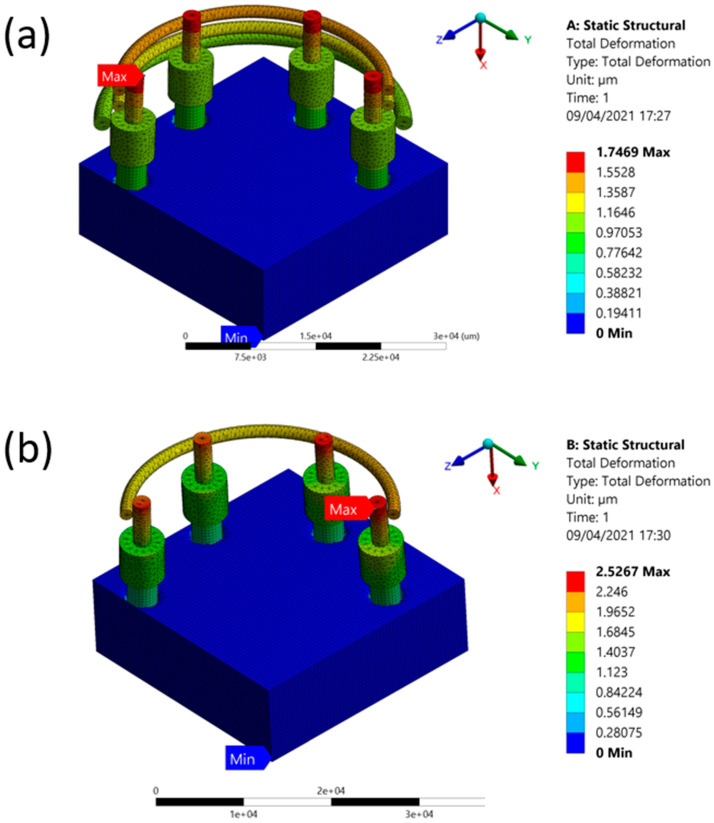
(**a**) Deformation when using three fences. (**b**) Deformation when using one fence.

**Figure 2 jcm-10-03986-f002:**
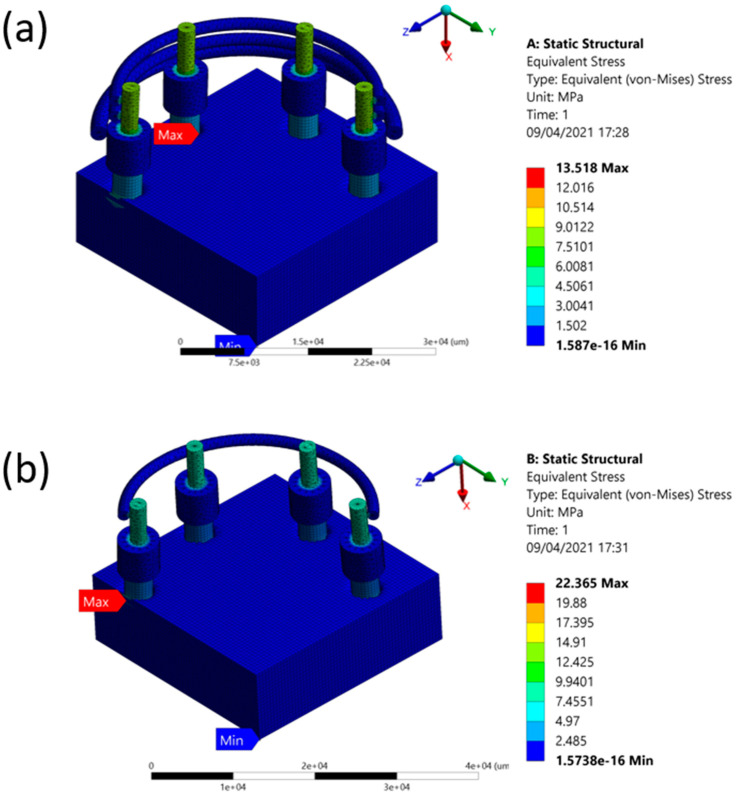
(**a**) Stress when using three fences. (**b**) Stress when using one fence.

**Figure 3 jcm-10-03986-f003:**
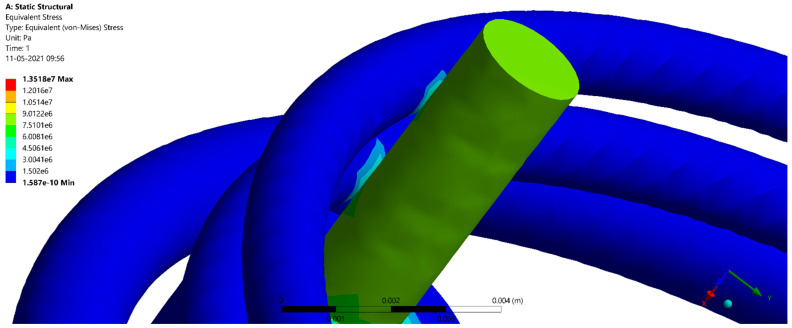
Stress dissipated along the fences.

**Figure 4 jcm-10-03986-f004:**
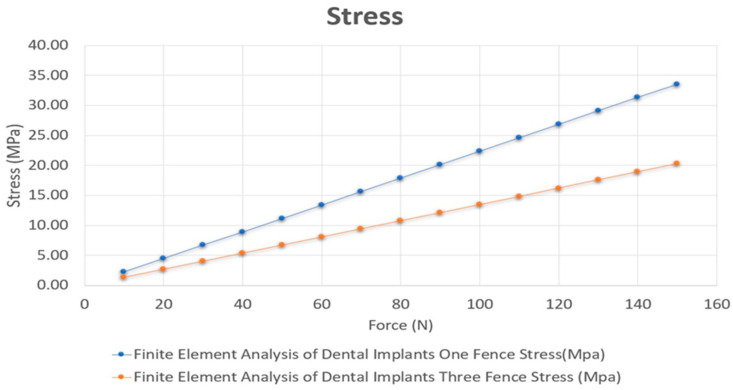
Deformation analysis of the two designs at various loads.

**Figure 5 jcm-10-03986-f005:**
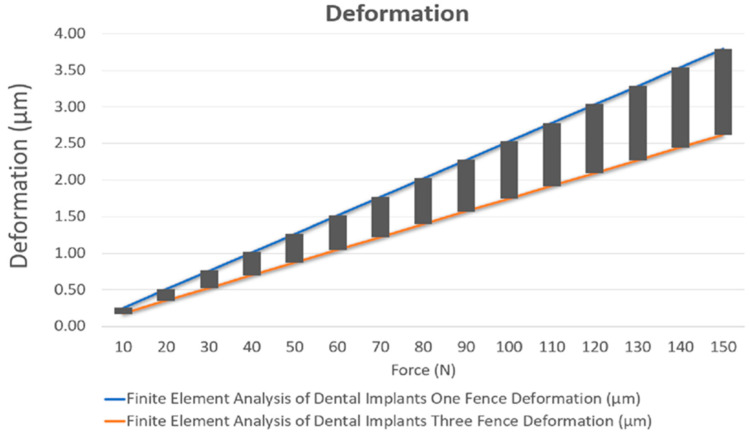
Stress analysis of the two designs at various loads.

**Figure 6 jcm-10-03986-f006:**
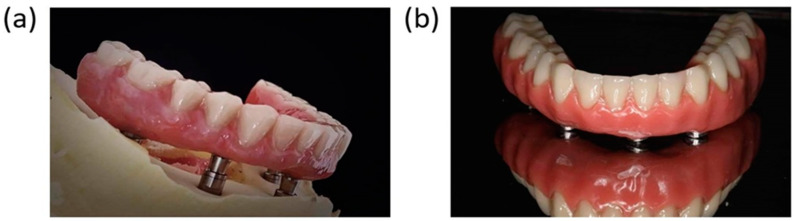
(**a**) Deformation when using three fences. (**b**) Deformation when using one fence.

**Table 1 jcm-10-03986-t001:** Mechanical Properties of the Titanium Alloy.

Material	Density (g/cm³)	Young’s Modulus (MPa)	Poisson’s Ratio	Compressive Yield Strength (MPa)
Titanium Alloy	4.620	96,000	0.36	930
Cancellous Bone	2.08	71,000	0.3	280
Cortical Bone	1.2	2000	0.3	112

**Table 2 jcm-10-03986-t002:** Deformation Values for Titanium Framework.

Object	Minimum (µm)	Maximum (µm)	Average (µm)
One Fence	0	2.5267	0.21521
Three Fences	0	1.7469	0.22412

**Table 3 jcm-10-03986-t003:** Stress Values for Titanium Framework.

Object	Minimum (MPa)	Maximum (MPa)	Average (MPa)
One Fence	1.5738 × 10^−16^	22.356	0.47612
Three Fences	1.587 × 10^−16^	13.518	0.46134

**Table 4 jcm-10-03986-t004:** Deformation Values for an Angular Load of 100 N.

Object	Minimum (µm)	Maximum (µm)	Average (µm)
One Fence	0	171.95	9.5416
Three Fences	0	169.04	11.36

**Table 5 jcm-10-03986-t005:** Stress Values for an Angular Load of 100 N.

Object	Minimum (MPa)	Maximum (MPa)	Average (MPa)
One Fence	4.5963 × 10^−18^	320.7	3.0263
Three Fences	1.1637 × 10^−16^	245.34	3.1472

**Table 6 jcm-10-03986-t006:** Deformation Values when Increasing the Vertical Load on the Posterior Jaw to 150 N while Decreasing the Load on the Anterior Jaw to 50 N.

Object	Minimum (µm)	Maximum (µm)	Average (µm)
One Fence	0	6.8344	0.42715
Three Fences	0	5.366	0.42004

**Table 7 jcm-10-03986-t007:** Stress Values when Increasing the Vertical Load on the Posterior Jaw to 150 N while Decreasing the Load on the Anterior Jaw to 50 N.

Object	Minimum (MPa)	Maximum (MPa)	Average (MPa)
One Fence	3.2548 × 10^−16^	64.467	0.94536
Three Fences	3.257 × 10^−16^	40.963	0.925

## Data Availability

Not applicable.

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
