# Peer review of "Comparative Analysis of Stress and Deformation between One-Fenced and Three-Fenced Dental Implants Using Finite Element Analysis"

_jcm, 2021, doi:10.3390/jcm10173986_

Round 1
Reviewer 1 Report
Comments to the Author
This manuscript describes a FEA study investigating the effects of increasing the number of fences within the implants that retain an All-in-four implant retained prosthesis. This is an important research area that can give clinicians valuable insight into the effects of selecting the best implant designs when restoring All-in-four prosthesis concepts. However, the current manuscript is not suitable for publication as is. There appears to be an issue with the methodology and not enough description of the study design. Moreover, the data were not appropriately presented in this manuscript. Specific comments are listed below.
- There is an issue concerning the methods used to analyze the data. There is no statical analysis conducted to examine the FEA results. Therefore, all calculations need to be redone, and results need to be generated and reported based on the correct statistical analysis.
- The authors never discuss the limitations of the study. They never describe on the paper the possible limitations with the FEA design and with the prosthesis selected, the distance between the implants. The connection between the implant and the prosthesis, implant angulation, etc. The possible impact of that and the type of bone (Cortical vs. Cancellous) effect on load and force distribution.
- The Results section of the Abstract should contain some numerical values of the results.
- The authors need to provide more details on how the data was validated in the Materials and Methods section, such as content validity, sample size, and pilot testing. Additionally, the authors should provide more info on the study design reasoning. For example, how the implants were loaded (90-degree forces vs. non-axial forces) All the variables in the study should be well defined and explained.
- Authors may consider exploring whether the use of different implant diameters and the angulation of the distal implants affect the stress distribution to make the paper more meaningful.
Reviewer 2 Report
Reagarding the litterature supporting this paper, I suggest the authors to consider also another factor which can affect the stress and the fatigue of dental implants.
So it is important to investigate also the neck design and for this reason I strongly reccomend to evaluate these two articles and cite them in the discussion section:
1:
Cosola S, Toti P, Babetto E, Covani U, Peñarrocha-Diago M, Peñarrocha-Oltra D. In-vitro fatigue and fracture performance of three different ferrulized implant connections used in fixed prosthesis. J Dent Sci. 2021 Jan;16(1):397-403. doi: 10.1016/j.jds.2020.08.002. Epub 2020 Aug 27.
2:
Cosola, S.; Toti, P.; Babetto, E.; Covani, U.; Peñarrocha-Diago, M.; Peñarrocha-Oltra, D. In-Vitro Investigation of Fatigue and Fracture Behavior of Transmucosal versus Submerged Bone Level Implants Used in Fixed Prosthesis. Appl. Sci. 2021,11, 6186. https://doi.org/10.3390/app11136186
Reviewer 3 Report
Specifics comments:
- Line 4: The authors are advised to replace “the specific objective” by “the aims”.
- Line 20: Please correct the word “The” at the beginning of the sentence.
- Line 22-27: The authors mentioned the difference in term of force distribution between implants and teeth. They are advised to add more literature concerning these statements. Additionally, it can be interesting for the readers to add the definition of osseointegration of Brånemark.
- Line 31-32: Same remark as previously, please add some references concerning the statements you mentioned.
- Line 40-43: The mention of the content of the section 2, 3 and 4 are unnecessary. The authors are advised to mention only the aims.
- Line 54: The use of “Biomaterial” can be a little bit confusing for the readers. The authors are advised to replace the title of this section by “Selection of implant material” or something similar.
- Line 65: “Clinical studies to prove the long-term success of zirconia implants are lacking”: this sentence is not correct as randomized clinical trials with a follow-up of 80 months have been published recently. Please revise this sentence and the authors are advised to update the references they used.
- Line 70: The last sentence is not clear, please rephrase.
- Line 77: Concerning “short implants” and due to the availability in the market of dental implant with a length between 4 and 7 mm, the appellation “short implants” must be reserved for implant with a length than seven millimeters. Please rephrase and modify this sentence.
- Line 80: Please remove the dot before the word “which”.
- Line 81: The authors are advised to replace “In dental implant” by “Concerning dental implants”.
- Line 105 to 111: This part must be added in the introduction part.
- Line 112-113: Please add details about the manufacturer of the software, the version etc…
- Line 117: What type of “Titanium alloy”?
- Table 2 and 3: The authors present the average of deformation and stress values. If you have an average, how many time did you calculated the deformation and stress values? This mention misses in the method part. Additionally, these results must be indicated as average +- standard deviation.
- Results part: A subsidiary question: why did you mentioned the figure 4 and 5 in the discussion part and not in the result part?
- Line 169: Please divide the sentence in two.
General comments:
- Interesting article concerning the simulation of stress and deformation between one-fenced and three-fenced dental implants.
- However, this article has some limitations. In the material and methods, the authors applied a masticatory force of 100N in a natural orientation along the x-axis. It is known yet that the occlusal biting load in the posterior jaw is usually about three times of that found in the anterior. Therefore, a specific section concerning the masticatory force must be insert and these forces have to be adjusted in the part “Material and Methods”. Concerning the method, how many time did you repeat the application of forces to evaluate the deformation and stress? Additionally, some precisions are missing in the “M&M” section.
- Moderate to extensive English changes are required.
Round 2
Reviewer 1 Report
Dear Authors,
Thank you for working on the clarifications and corrections to the paper.
Only minor editing to correct the English grammar and style is needed.
Best Regards
Author Response
We thank the reviewer for taking the time to review our paper and suggest valuble changes. The authors appreciate the comment of the reviewer. We would like to inform that the manuscript was edited for proper English language, grammar, punctuation, spelling,and overall style by native English-speaking editors at KGS Services (http://www.kgsupport.com). The revised manuscript was shown by tracking revisions.
Reviewer 3 Report
The authors revised the manuscript following the previous comments.
Extensive editing of English language was done.
Author Response
We thank the reviewer for taking the time to review our paper and suggest valuble changes.